# F-Doped Ni-Rich Layered Cathode Material with Improved Rate Performance for Lithium-Ion Batteries

Jinbo Zeng [1,2,3], Yue Shen [1,2,3,*], Xiufeng Ren [1,2], Xiang Li [1,2], Yanxia Sun [1,2], Guotai Zhang [1,2,3], Zhaowei Wu [1,2,3], Shenglong Zhu [1,2,3], Chunxi Hai [1,2] and Yuan Zhou [4,*]

1   Key Laboratory of Comprehensive and Highly Efficient Utilization of Salt Lake Resources, Qinghai Institute of Salt Lakes, Chinese Academy of Sciences, Xining 810008, China
2   Key Laboratory of Salt Lake Resources Chemistry of Qinghai Province, Xining 810008, China
3   University of Chinese Academy of Sciences, Beijing 100049, China
4   College of Materials and Chemistry & Chemical Engineering, Chengdu University of Technology, Chengdu 610059, China
*   Correspondence: shenyue@isl.ac.cn (Y.S.); zhouyuan1613@163.com (Y.Z.)

**Abstract:** Ni-rich layered cathode materials for lithium-ion batteries have received widespread attention due to their large capacity and low cost; however, the structural stability of the material needs to be improved. Herein, F-doped and undoped cathode materials prepared with an advanced co-precipitation method were used to measure the effect of F doping on the material. Compared to the undoped sample, the F-doped cathode materials exhibited an improved rate performance, because the porous structure of F-doped cathode materials is favorable for the infiltration of the electrolyte and the material, and the F-doped cathode material has a larger (003) crystal plane and a smaller Li$^+$ migration barrier energy. This simple F-doping treatment strategy provides a promising way to improve the performance of Ni-rich layered cathode materials for lithium-ion batteries.

**Keywords:** co-precipitation method; F-doped; Ni-rich layered cathode materials





## 1. Introduction

Ni-rich layered ternary transition metal oxides LiNi$_{0.8}$Co$_{0.1}$Mn$_{0.1}$O$_2$ have attracted increasing attention due to their high specific capacity of about 200 mAh g$^{-1}$, excellent rate capability, good cycle performance, and low cost, which makes them one of the best potential power battery cathode materials [1–10]. The preparation method for Ni-rich layered cathode material has a great influence on the microstructure and electrochemical properties of the material. The methods of preparing Ni-rich ternary layered cathode material are similar to that of an ordinary ternary cathode material, including the high temperature solid phase method [11,12], co-precipitation method [13,14], sol-gel method [15,16], spray drying method [17–19], combustion method [20–27], and so on. The solid-phase reaction method is a method that was conceived of earlier and is used less now, while the sol-gel method and the spray method have lower output and higher equipment requirements. The advanced co-precipitation method prepares a uniform spherical cathode material precursor, and then prepares a spherical cathode material with uniform particle distribution. However, the dissolution of transition metal ions; high solid electrolyte interface (SEI) layer impedance, Li$^+$/Ni$^{2+}$ mixing, transformation of layered structure to spinel and rock-salt structures, and the relatively poor cycle performance of Ni-rich ternary cathode materials seriously affects its large-scale application [7,28–31]. To overcome the above problems, various modification methods have been adopted, such as surface coating modification [1,5,32–37], preparation of concentration gradient distributed materials [3,36,38–40], doping modification [32,33,36,41–46], and so on. Compared with cation ion doping, anion doping, especially F-doping in Ni-rich cathode materials is less frequently reported [32,41,47]. Wang et al. [41]

found that when the F-doping amount was 4% of the weight of the precursor, the electrochemical performance of the cathode material was better, the capacity was 157.8 mAh g$^{-1}$ after 100 cycles at the current rate of 2 C, and the retention was 98.3%; the authors believe that Metal-F (M-F) bonds replace part of Metal-O (M-O) bonds, and M-F bonds better stabilize the structure of the material, prevent the surface of the material from being corroded by HF, and reduce the polarization of the material during cycling. Kim et al. [47] found that when the F-doping amount was 6 % of the cathode material, the capacity was 169.6 mAh g$^{-1}$ after 100 cycles at the current density of 100 mA g$^{-1}$, and the retention was 96.8%. Compared with the undoped material, the F-doped material provided better cycling performance, which the authors believed to be related to the strong M-F bonds and Li mobility enhancement, but excess F doping leads to Li$^+$/Ni$^{2+}$ mixing and worsens the material properties.

In this study, the commercial, widely used advanced co-precipitation method was applied because the pH and dosing rate are precisely controlled and the co-precipitation method can obtain precursors with Ni, Co, and Mn atoms distributed uniformly at the atomic level [48,49]. The F-doped Ni-rich cathode materials were prepared using a simple method; we found that a small amount of low-cost F-doping can lead to a large rate performance improvement, which is a cost-effective strategy to improve the performance of Ni-rich cathode materials for lithium-ion batteries.

## 2. Materials and Methods

### 2.1. Coprecipitation Synthesis of LiNi$_{0.8}$Co$_{0.1}$Mn$_{0.1}$O$_{2-x}$F$_x$ (x = 0.0005, 0.001, 0.005)

Advanced ternary transition metals coprecipitation was used to synthesize the Ni$_{0.8}$Co$_{0.1}$Mn$_{0.1}$(OH)$_2$ precursor. Analytical reagents (AR) NiSO$_4\cdot$6H$_2$O, CoSO$_4\cdot$7H$_2$O and MnSO$_4\cdot$H$_2$O were dissolved in deionized water with the stoichiometric proportion of 8:1:1 to form a 2 M transparent dark green transition metal sulfate saline aqueous mixture, AR NaOH was added to deionized water to create a 4 M alkaline aqueous solution, and an approximative 0.4 M NH$_4$OH solution made by adding AR NH$_3\cdot$H$_2$O to deionized water for use as a chelating agent. Firstly, a bottom liquid NH$_4$OH solution was added to a 50 L 306 stainless steel reaction with an agitator and a water bath temperature control. The agitator was set to a constant mixing speed of 600 rpm and the reactor temperature was set to 50 °C continuously, then the still was sealed and connected with protective N$_2$ gas airflow to protect transition metal ions from oxidation. Saline mixture aqueous, alkaline aqueous and NH$_4$OH solution were pumped into the still through sealed tubes by three precision peristaltic pumps synchronously, the velocities of flow of saline mixture aqueous and NH$_4$OH solution pumps were set to 40 mL min$^{-1}$, while the velocity of flow of alkaline aqueous pump was controlled by a precision pH meter to maintain pH = 11.1 ± 0.1. The coprecipitation reaction was carried out for 20 h, then the pumps were shut down and the water bath was cooled to pour out the suspension. The suspension then was washed with deionized water serval times with a centrifuge, and the deposit was dried in a vacuum drying oven overnight. Finally, the Ni$_{0.8}$Co$_{0.1}$Mn$_{0.1}$(OH)$_2$ precursor fine powder was obtained.

The Ni-rich layered LiNi$_{0.8}$Co$_{0.1}$Mn$_{0.1}$O$_2$ (NCM811) cathode material was synthesized by performing a solid-state reaction. Firstly, the Ni$_{0.8}$Co$_{0.1}$Mn$_{0.1}$(OH)$_2$ precursor was mixed evenly with battery-level nano LiOH·H$_2$O power with the molar ratio of 1:1.04, the excessive Li raw material was accounted for by Li evaporation in solid phase reaction. Secondly, the mixed powder was roasted in a tubular furnace with an O$_2$ flow of 2.0 L min$^{-1}$, the temperature schedule was 500 °C for 6 h and 805 °C for 12 h, followed by furnace cooling, and the heating rate was maintained at 5 °C min$^{-1}$. Finally, the roasted powder was grounded softly by an agate mortar and the as-prepared product was obtained. The F-doped fluorine gradient distributed Ni-rich layered LiNi$_{0.8}$Co$_{0.1}$Mn$_{0.1}$O$_{2-x}$F$_x$ (x = 0.0005, 0.001, 0.005) cathode material was synthesized by adding different amounts of NH$_4$F with molar ratios of NH$_4$F to Ni$_{0.8}$Co$_{0.1}$Mn$_{0.1}$(OH)$_2$ of 0.0005, 0.001, and 0.005 into the Ni$_{0.8}$Co$_{0.1}$Mn$_{0.1}$(OH)$_2$ precursor before mixing separately, corresponding to 500, 1000 and 5000 ppm. Thereafter,

the primitive Ni-rich layered $LiNi_{0.8}Co_{0.1}Mn_{0.1}O_2$ cathode material, and the F-doped Ni-rich layered $LiNi_{0.8}Co_{0.1}Mn_{0.1}O_{2-x}F_x$ (x = 0.0005, 0.001, 0.005) cathode material samples were named as F0, F500, F1000 and F5000, respectively.

## 2.2. Morphology and Structure Characterization

To identify the morphology and the structure, the precursor was observed using field emission scanning electron microscopy (SEM, Hitachi Su8010, Tokyo, Japan). Additionally, the as-prepared powders were observed under a field emission scanning electron microscope (FESEM, ZEISS Crossbeam 350, Jena, Germany) equipped with an EDS unit (EDS, OXFORD ULTIM MAX, Oxford, UK) and a high-resolution transmission electron microscope (HRTEM, JEOL JEM-2100F, Tokyo, Japan). Then, the precursor and the cathode materials were detected by an X-ray diffraction diffractometer with Cu-K$\alpha$ radiation (XRD, Bruker D8 Advance, Karlsruhe, Germany). The surface elemental changes were determined using an X-ray photoelectron spectroscopy spectrometer with Al-K$\alpha$ source (XPS, ThermoFisher ESCALAB 250Xi, Waltham, MA, USA). The elements contents were determined by a inductively coupled plasma-optical emission spectrometer (ICP-OES, PerkinElmer Avio500, Singapore, Singapore) and an ion chromatograph (IC, ThermoFisher Dionex ICS-6000 HPIC, Waltham, MA, USA).

## 2.3. Electrochemical Measurements

The cathode materials were then pulped and coated onto the collector to make the cathode. With a weight ratio of 8:1:1, the cathode materials, super P and polyvinylidene fluoride (PVDF) were well mixed in N-methyl pyrrolidone to make pulp; the pulp was then coated onto an aluminum foil evenly before drying, the coated aluminum foil was dried at 60 °C for 1 h and followed by 120 °C for 12 h, and it was then cooled in a vacuum drying oven. The dried coated aluminum foil was sliced to cathode disks with a diameter of 14 mm.

CR2025 coin type cells were assembled in a glove box filled with pure argon for electrochemical performance tests. The prepared cathode disks and lithium metal disks with diameters of 14.5 mm were used for the cathode and anode, separately. Celgard 2400 was used as the separator. Additionally, the electrolyte of 1 M $LiPF_6$ was dissolved in an equal-volume of diethyl carbonate (DEC) and ethylene carbonate (EC).

The cycling performance and rate performance of the cells were investigated on a battery testing system (Lanhe CT2001A, Wuhan, China), the galvanostatic charge–discharge was cycled between 2.8 and 4.3 V at different current rate (1 C corresponds to 170 mA $g^{-1}$) and the cyclic voltammetry (CV) and the alternating current (AC) electrochemical impedance spectra (EIS) of a frequency range from $10^5$ to $10^{-2}$ Hz were studied on an electrochemistry station (Ametek Princeton Applied Research PMC 1000, Berwyn, PA, USA). All the electrochemical measurements were carried out at round 25 °C in room.

## 3. Results

### 3.1. Structure and Morphology Analysis

The XRD pattern and SEM image of the $Ni_{0.8}Co_{0.1}Mn_{0.1}(OH)_2$ precursor synthesized by advanced ternary transition metals coprecipitation are displayed in Figure 1. The precursor is indexed to a hexagonal $Ni(OH)_2$ structure (ICSD#98-002-8101) with a P-3m1 space group. There is not any apparent impurity phase in the pattern which indicates the success of the coprecipitation synthesis and high purity of the precursor. The precursor powder is spheroid, and the primary particles are in a needle shape. The particle size of the secondary particles is about 5~12 μm.

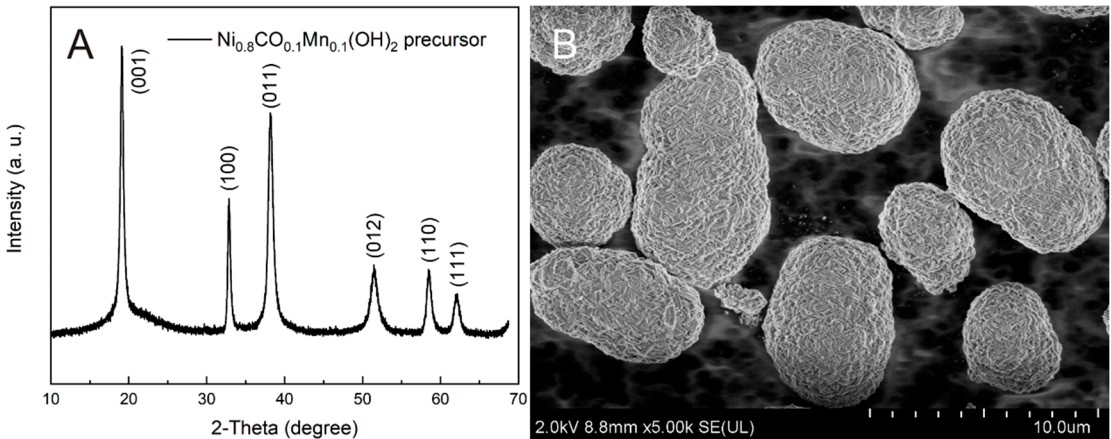

**Figure 1.** XRD pattern (**A**) and SEM image (**B**) of the $Ni_{0.8}Co_{0.1}Mn_{0.1}(OH)_2$ precursor.

After lithium mixing, doping and heat treatment, the cathode materials were obtained. The XRD patterns and related data are presented in Figure 2 and Table 1. The structures were matched to a hexagonal $\alpha$-NaFeO$_2$ structure with an R-3m space group. There are no apparent impurity phases in the patterns, which indicates the success of the synthesis, high purity of the cathode materials, and the success of F-entering the lattice of the cathode materials [47]. For each sample, the split of (006)/(012) and (018)/(110) is apparent. This indicates that each sample has a layered structure. Additionally, the I(003)/I(104) values of all the samples are higher than 1.2 which indicates that the samples have low $Li^+$/$Ni^{2+}$ cation mixing [47,50]. The F5000 sample has a lower I(003)/I(104) value than the other samples, which indicates that F5000 has relative higher cation mixing. A higher concentration of F-doping may cause higher cation mixing, and lead to a poorer electrochemical performance.

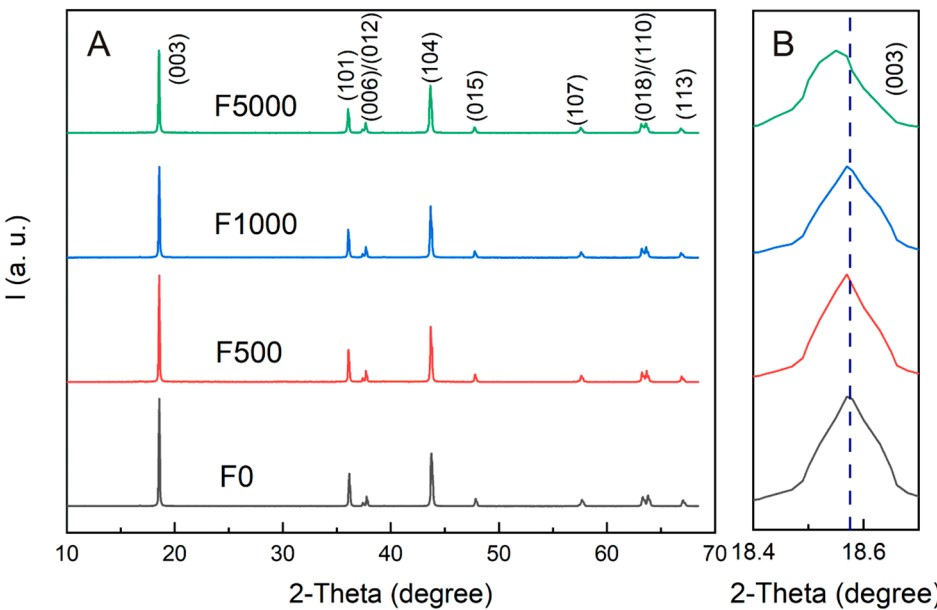

**Figure 2.** XRD patterns (**A**) and partial magnification around 18.5 degree (**B**) of the samples F0, F500, F1000 and F5000.

**Table 1.** The chemical content and XRD patterns data of the samples F0, F500, F1000 and F5000.

| Sample | I(003) (Counts) | I(003)/I(104) | Molar Ration of Li:Ni:Co:Mn | Molar Percent of F |
|---|---|---|---|---|
| F0 | 22057 | 2.0661 | 1.01:0.79:0.099:0.098 | 0 |
| F500 | 21427 | 1.9120 | 1.01:0.80:0.098:0.099 | 0.049% |
| F1000 | 18490 | 1.7762 | 1.01:0.80:0.099:0.098 | 0.10% |
| F5000 | 16675 | 1.7271 | 1.01:0.80:0.099:0.099 | 0.51% |

It can be noted that the intensity of (003) peaks and I(003)/I(104) values reduce, and (003) peaks shift to a lower angle with the increase in the F-doping amount. The lower angle of the (003) peak indicates an increment of the (003) interplanar distance, because more transition metal ions are in high valence states induced by electrical neutrality compensation after F-doping [47]. A larger (003) interplanar distance may result in a lower barrier energy of lithium-ion de-intercalation and acculturate the Li$^+$ migration rate. The chemical contents were determined by ICP-OES and IC tests, which are listed in Table 1, and matched the designed molar rations well.

The Rietveld refinement results of XRD patterns for all samples are presented in Figure 3 and Table 2. All the Rietveld refinement weighted profiles residual errors (R$_{wp}$) were less than 15%; therefore, it can be concluded that the refined results are reasonable [51]. Notably, the a and c cell parameters of all the samples increased with the increase in F doping, and the Li-F bond was stronger, resulting in a repulsive force in the oxide system and a larger unit cell [41]. A larger c cell parameter brings a larger Li$^+$ diffusion channel, and may lead to a better rate performance, and a larger $c/a$ value means a more ordered hexagonal layered structure [23]. However, excessive F-doping will aggravate the mixing of Li$^+$ and Ni$^{2+}$, which will eventually lead to the degradation of the electrochemical performance of the material [23,41,47]. The $c/a$ values of all the samples are larger than 4.9, which indicates that all the samples have good electrochemical performance.

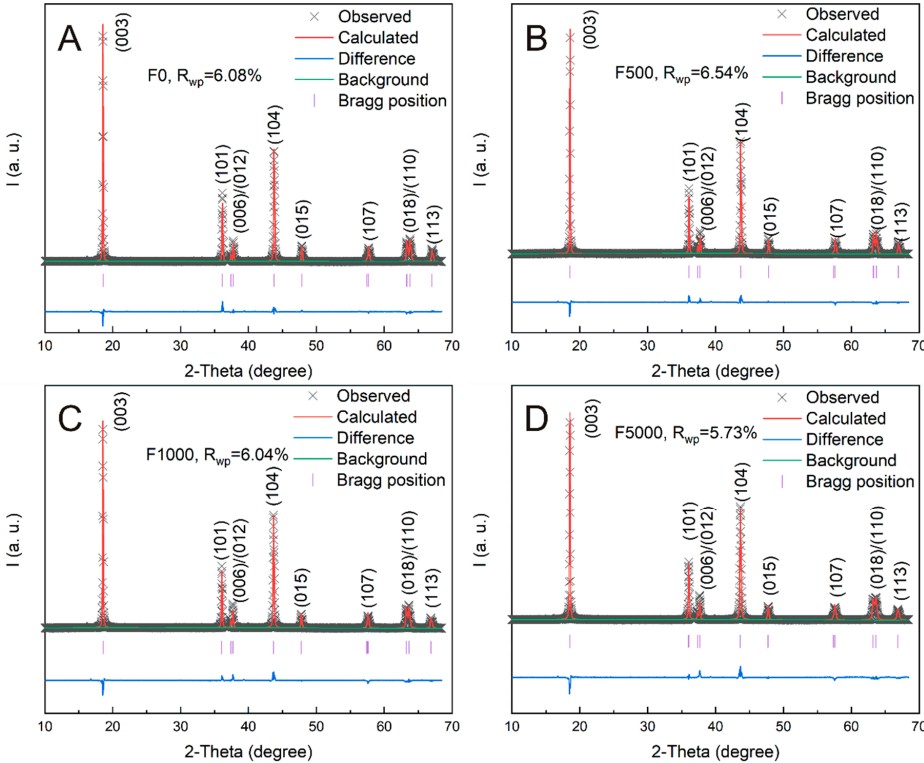

**Figure 3.** XRD patterns Rietveld refinement results of the samples F0 (**A**), F500 (**B**), F1000 (**C**) and F5000 (**D**).

**Table 2.** The lattice constants and I(003)/I(104) of XRD patterns of the samples F0, F500, F1000 and F5000.

| Sample | $a$ (Å) | $c$ (Å) | $c/a$ |
|--------|---------|---------|-------|
| F0 | 2.8679 | 14.1701 | 4.9409 |
| F500 | 2.8741 | 14.2023 | 4.9415 |
| F1000 | 2.8773 | 14.2418 | 4.9497 |
| F5000 | 2.8783 | 14.2433 | 4.9485 |

To study the F-doping influence on the morphology and the chemical content of the samples, FIBSEM and EDS lining and mapping measures were conducted. The SEM and FIBSEM images are presented in Figure 4 and EDS lining and mapping results are displayed in Figure 5. Obviously, the secondary particle size of as-prepared cathode material after heat treatment is similar to the secondary particle size of the precursor, however, the shape of the primary particles changed from the needle shape of the precursor to the polyhedron shape of the cathode material. Additionally, the internal porosity of the secondary particles for each sample increased with the increase in the F-doping amount. This means the introduction of F results in a certain degree of etching on the surface of the primary particles of the material, which further reduces the crystallinity of the material. This is consistent with previous XRD results. Such a morphology is helpful for the infiltration of the electrolyte and shortens the deintercalation channel of Li, which is beneficial for improving the rate performance of the material, but on the other hand, it also accelerates the side reaction between the electrolyte and the primary particles. Furthermore, we found the oxygen and fluorine gradient distributed structures of the cathode materials in EDS lining and mapping measurements. For comparison, the undoped sample F0 (Figure 5A) and the F-doped sample F500 (Figure 5B) were selected. Clearly, for the undoped F0, it can be seen from the line distribution (the purple line of O Kα1 in Figure 5A) and surface distribution (the red mapping image of O Kα1 in Figure 5A) of the oxygen of the sample that the surface oxygen content was higher than the inner oxygen content, and there was a gradient distribution from the surface to the interior more specifically. Furthermore, for the F-doped F500, F (the brown line and the red mapping image in Figure 5B) and O (the purple line and the green mapping image in Figure 5B) are gradient distributed from the surface to the interior. The high content of O on the surface may be related to the fact that the cathode material is processed in an oxygen atmosphere, and the high content of F on the surface may be caused by the fact that the precursor was soaked in the $NH_4F$ aqueous solution before heat treatment, The infiltration of the precursor in the $NH_4F$ aqueous solution results in a higher F content on the surface of the precursor than in the interior.

To further analyze the F-doping influence on the morphology and the structure of the materials, HRTEM and SAED measurements were carried out, for which the results are presented in Figure 6. The lattice fringes with an interplanar spacing of 0.473 nm in Figure 6A correspond to the (003) planes of the undoped F0. Two suits of the selected area electron diffraction spots belong to the (110) and (101) planes of the undoped F0 which are presented in Figure 6A1. Contrastingly, the lattice fringes with interplanar spacing of 0.474 nm in Figure 6B correspond to the (003) planes of the F-doped F500. Three suits of selected area electron diffraction spots belong to the (110), (104) and (101) planes of the undoped F0 which are presented in Figure 6B1. Noticeably, the lattice fringes corresponding to the (003) planes of the F-doped F500 are wider than those of the undoped F0, which is consistent with previous XRD results. Furthermore, the lattice fringes for the F-doped F500 sample are blurrier than those of the undoped F0 and the selected area electron diffraction spots for the F-doped F500 are circles, blurry lattice fringes and selected area electron diffraction spots circles which indicate poor crystallinity and polymorphism, which proved the previous conclusion once again.

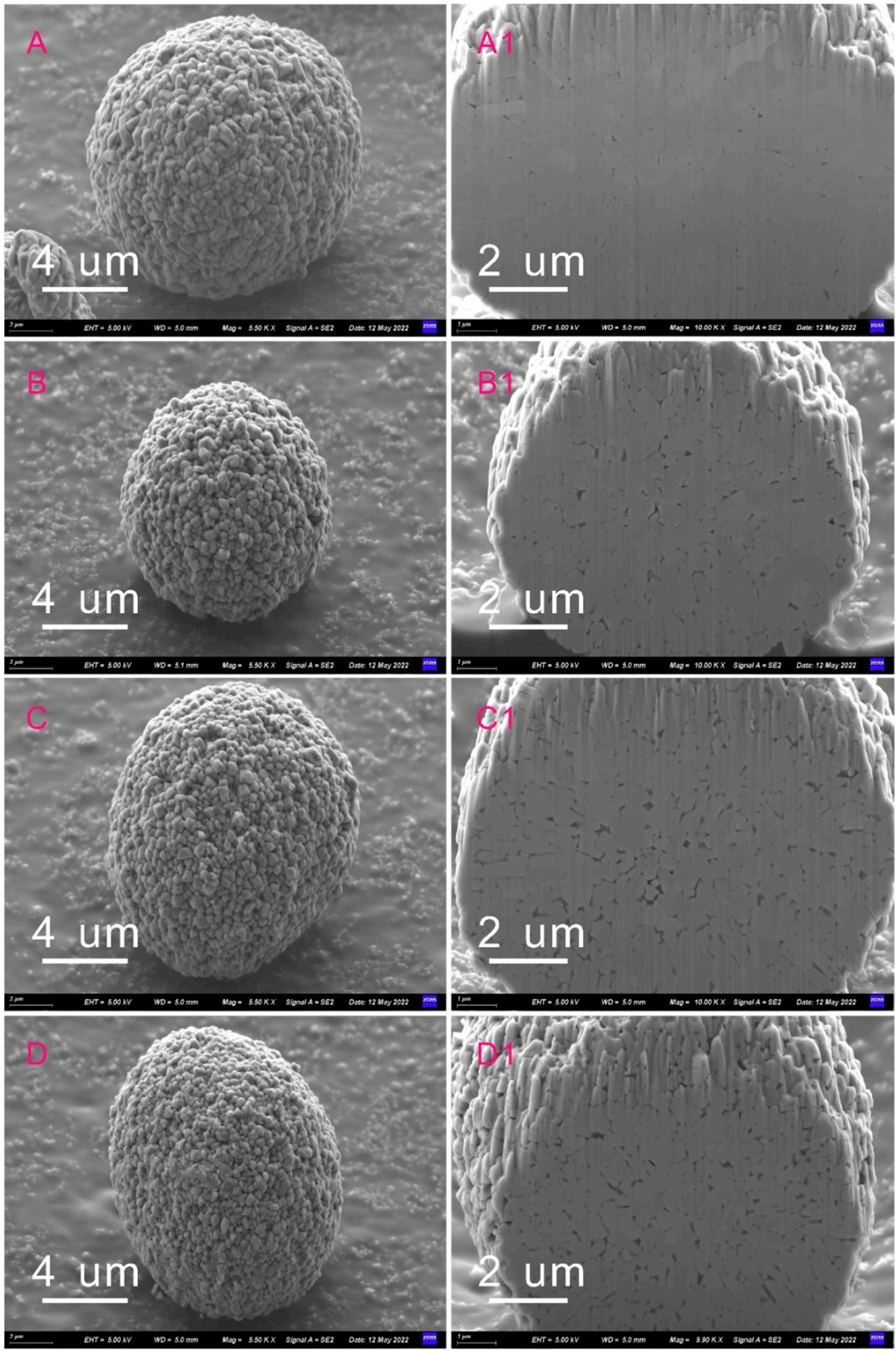

**Figure 4.** SEM and FIBSEM images of the samples F0 (**A**,**A1**), F500 (**B**,**B1**), F1000 (**C**,**C1**) and F5000 (**D**,**D1**).

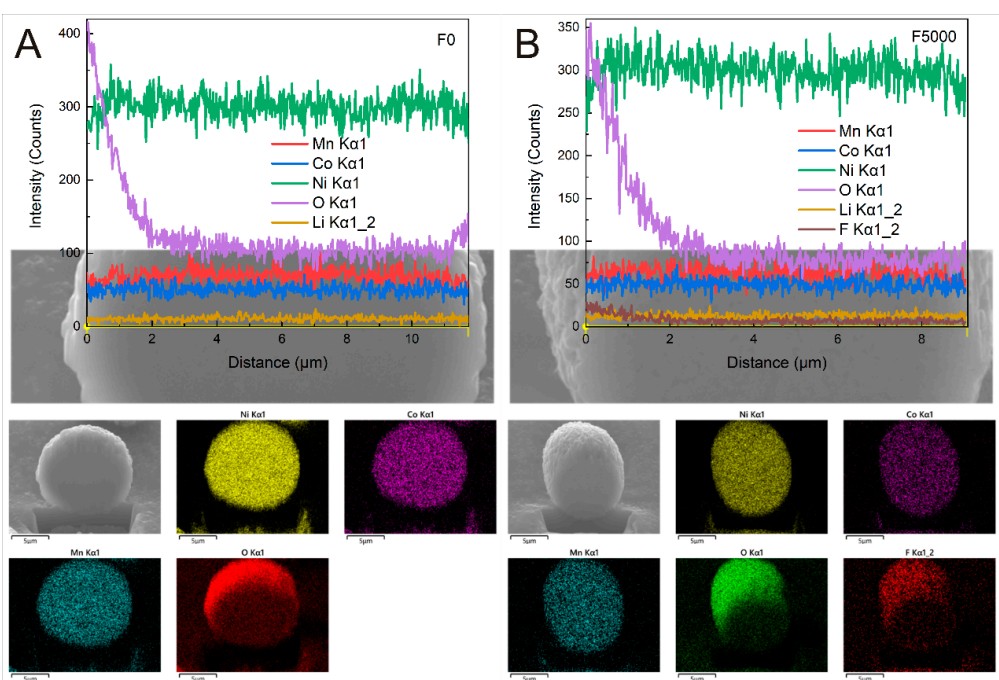

**Figure 5.** EDS lining and mapping results of the samples F0 (**A**) and F500 (**B**).

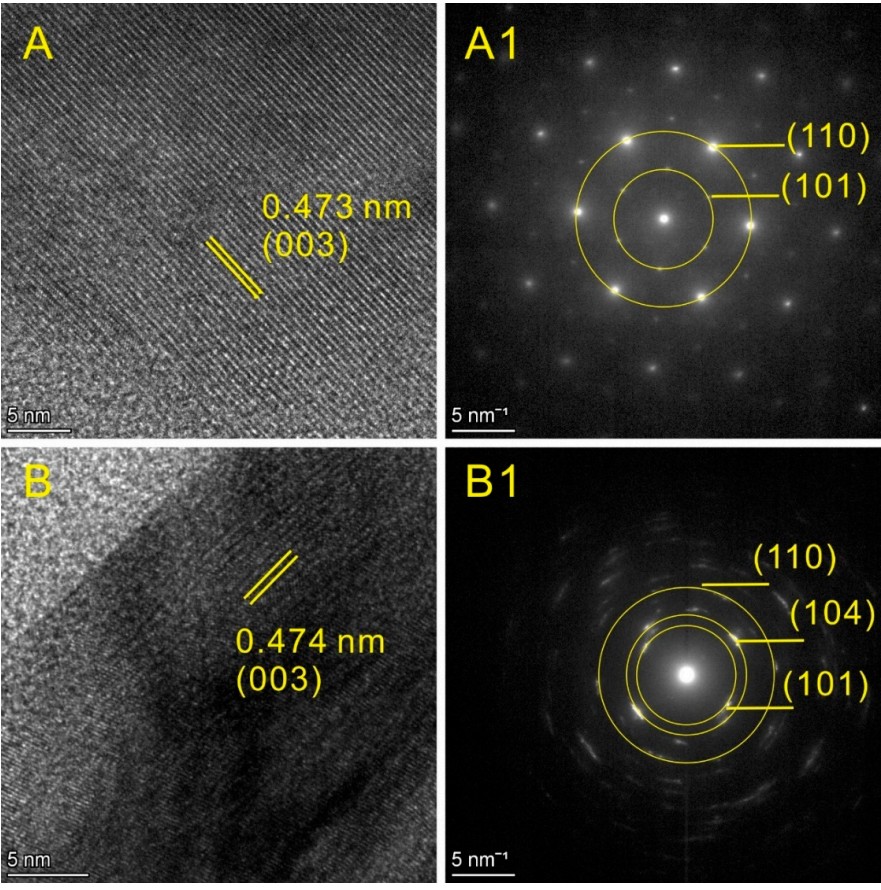

**Figure 6.** HRTEM and SAED results of the samples F0 (**A**,**A1**) and F500 (**B**,**B1**).

### 3.2. Electrochemical Performance

To evaluate the electrochemical performance of the materials, a cyclic voltammetry analysis was first performed with inactivated cells, for which the results are presented in Figure 7. With the increase in the F doping amount, the potential differences of the major oxidation and reduction peaks for the initial ($\Delta V_1$) and the second ($\Delta V_2$) cycle become larger, oxidation peaks move to a higher potential, and oxidation peaks around 4.0 V become less pronounced. Larger oxidation and reduction peaks separation indicates increased polarization of the electrode and poor electrochemical performance, and higher potential redox peaks lead to higher capacity [52].

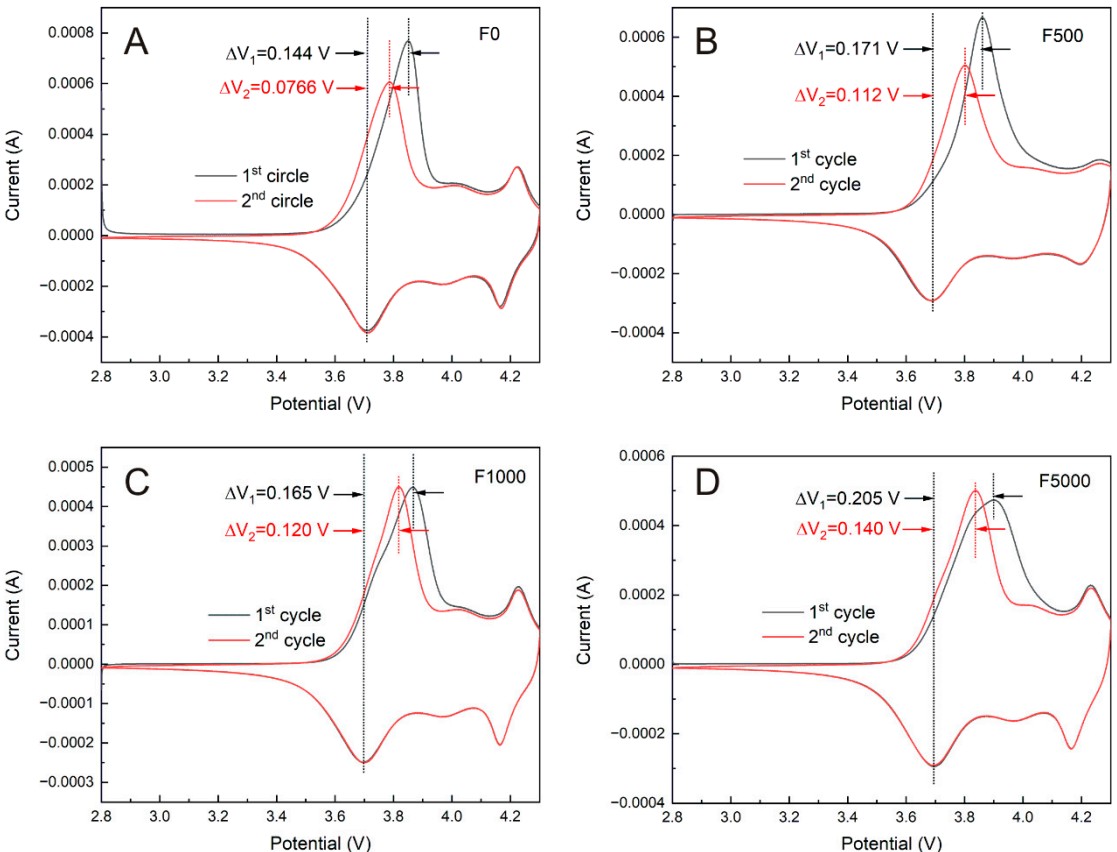

**Figure 7.** CV curves of the samples F0 (**A**), F500 (**B**), F1000 (**C**) and F5000 (**D**).

After three cycles of galvanostatic charge–discharge activation at a 0.1 C rate within 2.8 V to 4.3 V, galvanostatic charge–discharge was conducted to measure the rate capability and cycling performance of cells, for which the corresponding results are displayed in Figure 8. For the rate capability study, the cells of the samples F0, F500, F1000 and F5000 were charged and discharged for 5 cycles from 2.8 to 4.3 V at different rates of 0.5 C, 1 C, 2 C, 5 C, 8 C and 0.5 C. As can be seen in Figure 8A, for 5 cycles at 0.5 C, the average capacities for the samples F0, F5000, F1000 and F500 were 177.54, 182.12, 190.14 and 205.02 mAh $g^{-1}$, respectively, which are better than the related literature results [41,47]. For 10 cycles at 1 C and 2 C, similar results were achieved. For 10 cycles at high rates of 5 C and 8 C, the average capacities increase in the order of the F5000, F1000, F0 and F500 samples, the capacity of the F500 and the F5000 at 8 C are 157.76 and 128.44 mAh $g^{-1}$, which means a high amount of F doping is not beneficial for capacity at a high rate. When the charge–discharge current rate is back to 0.5 C, the F500 has a higher capacity than other samples. Therefore, this indicates that the F doping amount of 500 ppm is more suitable.

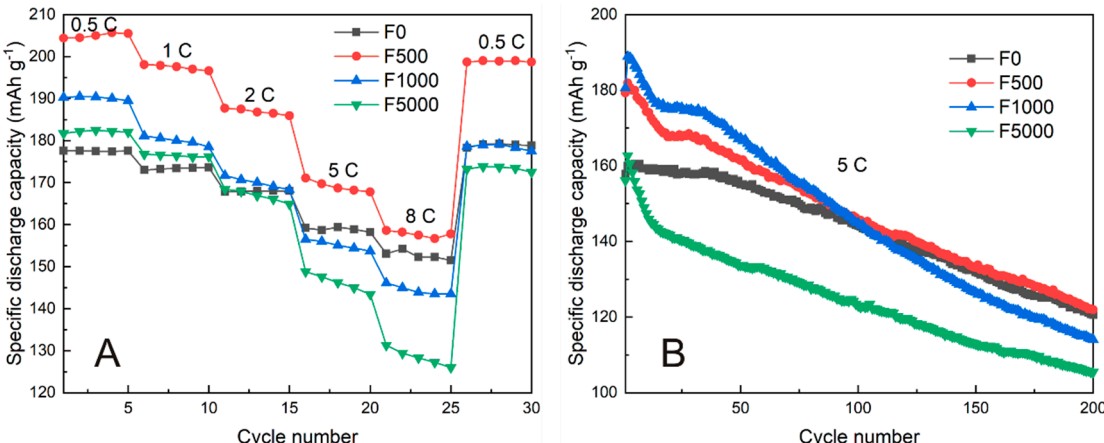

**Figure 8.** Rate (**A**) and cyclic (**B**) performance of the samples F0, F500, F1000 and F5000.

After three cycles at 0.1 C, the cyclic performance tests were carried out at a relatively high rate of 5 C for 200 galvanostatic charge–discharge cycles in the range of 2.8 to 4.3 V. From Figure 8B, we found that for the first 25 cycles the capacities of the F-doped samples drop quickly, and for the first 75 cycles the F-doped samples F500 and F1000 have higher capacities than others, but with further cycling, their capacities drop faster than the undoped samples. After 200 cycles, the capacities of samples F0, F500, F1000 and F5000 drop from 157.90, 179.30, 180.60 and 156.1 mAh g$^{-1}$ to 120.60, 122.00, 114.00 and 105.50 mAh g$^{-1}$, with a corresponding capacity retention of 76.38%, 68.04%, 63.12% and 67.58%, respectively. It can be seen that suitable F amount doping can increase the (003) interplanar spacing, improve Li-ion mobility, and enhance the performance of the material at high rates, no matter whether the amount of doping is more or less, F doping does not improve the cycle performance of the material, which is not consistent with the relevant literature conclusions, which may be related to the different systems of the respective materials [41,47].

*3.3. Density Functional Theory (DFT) Calculation*

To study the migration barrier energy of lithium ions in the undoped and F-doped Ni-rich materials, density functional theory (DFT) calculation using Cambridge Sequential Total Energy Package [53] (CASTEP 19.11) was carried out. The Perdew Burke Ernzerhof (PBE) form of the global gradient approximation was used. A cut-off energy of 600 eV was applied in the calculations, the electronic energy of the supercell was converged to $10^{-6}$ eV, and the force on all unconstrained atoms were converged to 0.02 eV Å$^{-1}$. The Monkhorst–Pack k-point grid was applied at $3 \times 3 \times 2$. Spin polarization was considered in all the calculations of this work. The transition state (TS) search jobs using the Linear Synchronous Transit (LST) and the Quadratic Synchronous Transit (QST) methods were used to determine the migration barrier energy. A $4 \times 4$ supercell of the conventional standard LiNiO$_2$ cell was used for further modeling, and 10 Ni ions were substituted by 5 Mn ions and 5 Co ions randomly. The effect U values of 2.5 eV, 2.5 eV and 2.5 eV were applied, respectively, for the Ni 3d, Co 3d and Mn 3d states, the chemical formula was Li$_{48}$Ni$_{38}$Co$_5$Mn$_5$O$_{96}$ after substitution which can be reduced to LiNi$_{0.792}$CO$_{0.104}$Mn$_{0.104}$O$_2$ and is similar to LiNi$_{0.8}$Co$_{0.1}$Mn$_{0.1}$O$_2$ (NCM811). After structure optimization, the undoped (Figure 9A) and F-doped (Figure 9B) cathode material models for DFT calculation in chemical formula Li$_{47}$Ni$_{38}$Co$_5$Mn$_5$O$_{96}$ and Li$_{47}$Ni$_{38}$Co$_5$Mn$_5$O$_{95}$F were used for the transition state search and the barrier energy calculation. This chemical formula is equivalent to a doping amount of F of 20000 ppm. The Li$^+$ migration path of this model is oxygen dumbbell hopping (ODH), which is the migration path of Li$^+$ in the early stage of charging [54]. For the undoped version, the barrier from the reactant is 0.65256 eV and the barrier from the product is 0.90033 eV, while for the F-doped version, the barrier from the reactant is 0.00536 eV and the barrier from product is 0.48930 eV. With the higher value as the migration energy, the barrier for the undoped version is 0.90033 eV, and the barrier for the F-doped version is

0.48930 eV. This shows that the doping of F can reduce the migration energy of Li$^+$ at the initial stage of charging, it represents a good rate performance and is consistent with the previous results and inferences.

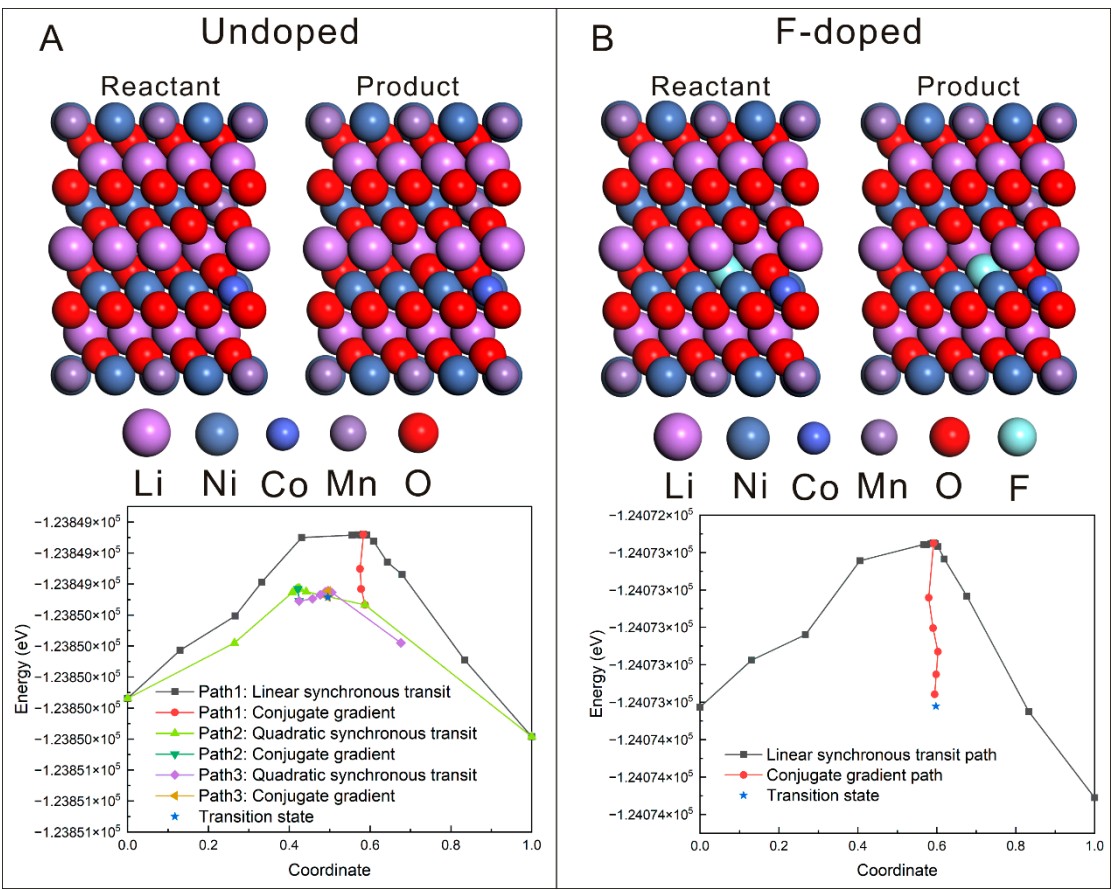

**Figure 9.** Models and transition state search result of the undoped (**A**) and F-doped (**B**) materials.

## 4. Discussion

### 4.1. EIS Analysis before and after Cycling

In order to investigate the effect of F-doping on the electrochemical kinetics of all the samples, EIS analysis on the cells before and after cycling was performed on the corresponding measured Nyquist plots, fitted Nyquist plots and the equivalent circuits used for fitting, which are illustrated in Figure 10 and the related results are presented in Table 3. The intercept at the $Z_{re}$-axis in the high frequency region reflects the ohmic resistance of the electrode ($R_E$), and the measured Nyquist plots have the following three parts: the semicircle in the high-to-medium frequency region which reflects the solid electrolyte interface resistance ($R_{SEI}$), the semicircle in the medium-to-low frequency region which reflects the charge transfer resistance ($R_{CT}$) and the linear plot in the low frequency region which reflects the Warburg impedance ($W_0$) [1,7,47,55–58]. D means the diffusion coefficient of Li$^+$ ions, and it can be calculated by Equation (1):

$$D = \frac{R^2 T^2}{2A^2 n^4 F^4 C^4 \sigma^2} \tag{1}$$

where $R$ = 8.314 J K$^{-1}$ mol$^{-1}$, is the gas constant; $T$ = 298.15 K, is the temperature; $A$ = 1.654 cm$^2$, is the area of the electrode; $n$ = 1, is the number of electrons in reaction; $F$ is

Faraday's constant 96485.34 C mol$^{-1}$; $C$ is the concentration of Li$^+$ ions 0.00718 mol cm$^{-3}$; and $\sigma$ is the Warburg factor. While the $\sigma$ can be confirmed by Equation (2):

$$Z_{re} = R_E + R_{SEI} + R_{CT} + \frac{\sigma}{\omega^{1/2}} \tag{2}$$

where $\omega$ is the angular velocity which equals 2πf, and f is the frequency of the Nyquist plot, the $\sigma$ is the slope of linear fit equation of the above equation [1,41,47,59]. By comparing the EIS analysis results of the cells before and after 200 cycles at 5 C from 2.8 to 4.3 V, it is found that the $R_{SEI}$ of the undoped sample F0 has a small increase after the cycle, while the $R_{SEI}$ of the F-doped sample becomes smaller after the cycle. This may be due to the fact that F doping can reduce the residual lithium on the surface of the material. However, the F-doped sample has a greater $R_{CT}$ than the undoped sample after the cycle, which may be due to the more severe side reactions between the electrolyte and the F-doped material [7]. The ohmic resistance of the electrode $R_E$ does not change much before and after cycling, and has little effect on the properties of the material. The diffusion coefficient of Li$^+$ ions (D) of the F-doped material is in the same order of magnitude as that of the undoped material, but the D of the F-doped material changes less before and after cycling.

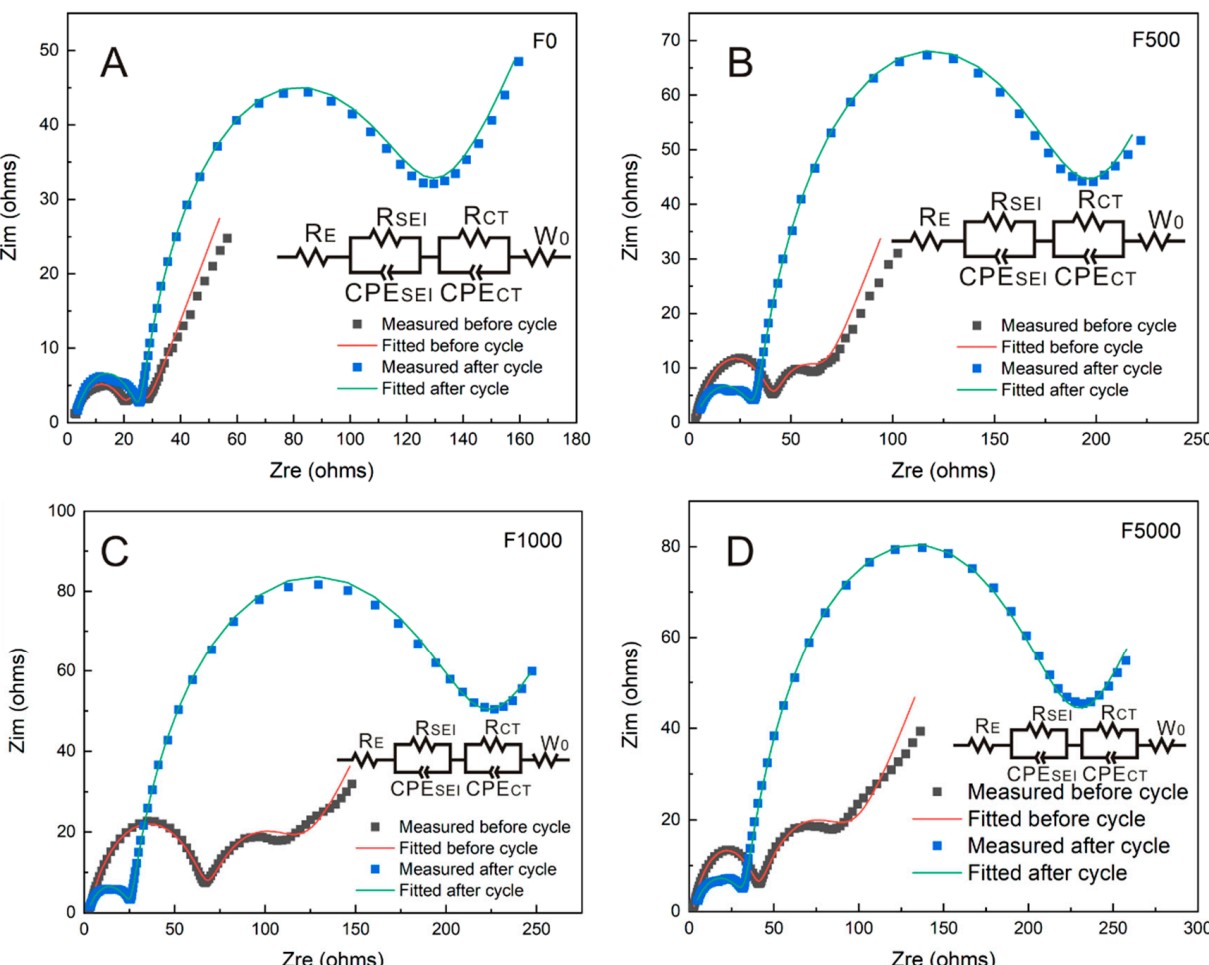

**Figure 10.** Nyquist plots and equivalent circuit before and after cyclic performance test of the samples F0 (**A**), F500 (**B**), F1000 (**C**) and F5000 (**D**).

**Table 3.** The EIS analysis data of the samples F0, F500, F1000 and F5000 before and after cyclic performance measurement.

| Sample | $R_E$ (ohms) | | $R_{SEI}$ (ohms) | | $R_{CT}$ (ohms) | | $D$ (cm$^2$ s$^{-1}$) | |
|---|---|---|---|---|---|---|---|---|
| | Before | After | Before | After | Before | After | Before | After |
| F0 | 1.76 | 2.08 | 19.84 | 23.12 | 4.26 | 87.89 | $3.81 \times 10^{-12}$ | $1.60 \times 10^{-12}$ |
| F500 | 2.73 | 2.17 | 38.97 | 30.63 | 19.16 | 141.30 | $1.04 \times 10^{-12}$ | $1.00 \times 10^{-12}$ |
| F1000 | 2.72 | 1.94 | 64.84 | 23.51 | 45.90 | 171.30 | $3.67 \times 10^{-12}$ | $1.28 \times 10^{-12}$ |
| F5000 | 2.34 | 2.32 | 47.92 | 30.88 | 38.15 | 174.20 | $1.85 \times 10^{-12}$ | $1.25 \times 10^{-12}$ |

*4.2. XPS Analysis before and after Cycling*

To investigate the deeper causes, XPS analysis on the cathode disks before and after cycling was carried out. The XPS spectra of the fresh and cycled samples are given in Figure 11. In the F 1s spectrum of the fresh undoped sample F0, there is only one peak around 687.80 eV which belongs to PVDF [23]. No metal-fluoride (M-F) bond peak around 685 eV was found, which is because the material was not doped with F. After 200 cycles at 5 C, the M-F peak around 685.43 eV was found, which is due to the appearance of LiF on the surface of the material during cycling [23]. The M-F peak around 685 eV was found on the surface of fresh F-doped sample F500, which further evidences that doping was successful, and the M-F peak of the cycled F-doped sample F500 becomes stronger. From Table 4, it can be seen that after cycling, the proportion of M-F in all F bonds is 63.21% for the undoped F0 and 35.90% for the F-doped F500. With the progress of the cycle, the M-F of the undoped sample increases, while the M-F of the doped sample increases slowly, which may be the reason for the decrease in $R_{SEI}$ for the doped sample after cycling in the above EIS analysis. The binding energy around 531 eV belongs to the lattice O on the surface of samples, while the binding energy around 532 eV belongs to the surface O of byproducts such as Li$_2$CO$_3$ [5,23,60]. After cycling, from the fact that there are many O peaks, we can see that there are many kinds of surface oxides generated by side reactions on the surface of the undoped F0 [23,41]. Compared to the undoped F0, the percent of lattice oxygen of the F-doped F500 falls faster; this shows that the structure of the layered cathode material is destroyed faster, which is why the $R_{CT}$ of the doped material increases rapidly and the cyclability decreases rapidly [23,41].

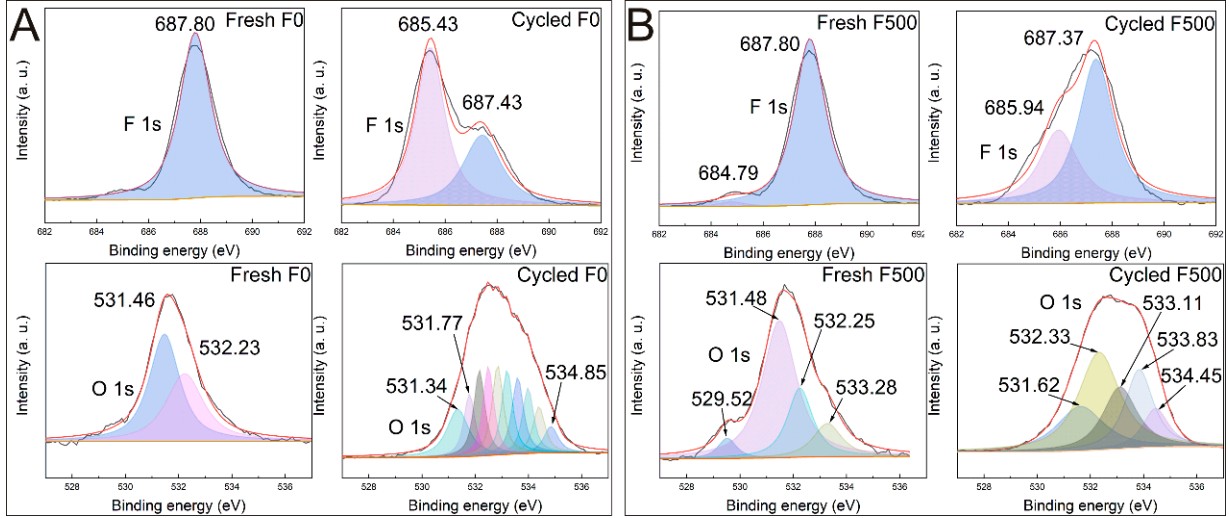

**Figure 11.** XPS spectra of the fresh and cycled samples F0 (**A**) and F500 (**B**).

**Table 4.** Surface concentration in atom percent of different elements of the fresh and cycled samples F0 and F500.

| Sample | M-F | | Lattice O | |
|---|---|---|---|---|
| | Before | After | Before | After |
| F0 | 0 | 63.21% | 56.53% | 12.66% |
| F500 | 2.27% | 35.90% | 65.20% | 13.98% |

## 5. Conclusions

This study successfully prepared an F-doped Ni-rich layered cathode material, of which a gradient distribution of O and F and a porous structure were found, and an improvement was achieved in rate performance for the suitable F amount doped sample, which is due to the fact that the porous structure facilitates the infiltration of electrolyte and shortens the deintercalation path of Li$^+$. Additionally, the F doping can increase (003) interplanar spacing and reduces the Li$^+$ migration energy. However, the cycle performance did not improve, because F-doping leads to a relatively lower crystallinity and higher Li$^+$/Ni$^{2+}$ mixing. A lower crystallinity and porous structure can help the infiltration of the electrolyte, there are more side reactions between the electrolyte and the F-doped cathode material, and the lattice oxygen consumption of the doped sample is faster. This destroys the layered structure, introduces a higher charge transfer resistance, and reduces cycling stability.

**Author Contributions:** Conceptualization, J.Z., Y.S. (Yue Shen), X.R., X.L., Y.S. (Yanxia Sun), G.Z., Z.W., S.Z. and C.H.; methodology, J.Z. and Y.S. (Yue Shen); software, J.Z. and Y.S. (Yue Shen); validation, J.Z. and Y.S. (Yue Shen); formal analysis, S.Z.; investigation, G.Z.; resources, Y.S. (Yue Shen), C.H. and Y.Z.; data curation, J.Z.; writing—original draft preparation, J.Z.; writing—review and editing, J.Z.; visualization, J.Z.; supervision, Y.S. (Yue Shen) and Y.Z.; project administration, Y.S. (Yue Shen); funding acquisition, Y.S. (Yue Shen). All authors have read and agreed to the published version of the manuscript.

**Funding:** This work was supported by the National Natural Science Foundation of China (NSFC No. 11904374), the CAS "Light of West China" Program, the Qinghai Provincial Talents Program for High-Level Innovative Professionals, the project for innovation platform construction of Qinghai Provincial Key Laboratory of Salt Lake resources chemistry (No. 2022-ZJ-Y06), the project for international cooperation in science and technology of Qinghai province (2021-HZ-810) and the special project for innovation platform construction of Qinghai Salt Lake resource chemistry key laboratory (2022-ZJ-Y06).

**Informed Consent Statement:** Informed consent was obtained from all subjects involved in the study.

**Data Availability Statement:** Not applicable.

**Conflicts of Interest:** The authors declare no conflict of interest.

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
