# Peer review of "F-Doped Ni-Rich Layered Cathode Material with Improved Rate Performance for Lithium-Ion Batteries"

_processes, doi:10.3390/pr10081573_

Round 1

Reviewer 1 Report

Dear Autor, 

your manuscripte describes the impact of F-doping amount on the performance of Ni-rich cathode materials for lithium-ion batteries. You showed that low F-doping can improve the performance at nearly constant cycle performance than without doping. It's a nice work with a very interesting contribution to the battery research community. 

Note:

+ line 170 and 250/251: missing blank 1) before the reference [51] and 2) between 2 C and 5 C.

+ verification/specification of statements, e.g. line 173: "... larger c cell parameter brings a larger Li+ diffusion ...", what does "larger" mean, how much larger. Further statements which need a verification are

- line 178: "... good electrocemical ..."

- line 242: "... becomes smaller, ..."

- line 262: "... sample drop quickly, ..."

- line 263: "... have higher capacities ..."

- line 331: "... becomes smaller after ..."

Kind regards, 

the Reviewer

Reviewer 2 Report

The writing of this paper is excellent. The manuscript contains some interesting achievements, and the results are well supported with sufficient discussions. However, some issues need to be seriously addressed before its acceptance and publication, as listed below:

1.      The authors should carefully elaborate the advantages of this study compared to the previous papers in literature. Make a comparison table of your work with previous reported literature.

2.      Is the XRD diagrams coherent with doped content?

3.      Please present dQ/dV curves for C/D profiles during cycling in Figure 8.

4.      Add capacity graph in Figure 8.

5.      Add the improved rate performance capacity in Abstract as well as in Conclusion.

6.      There are several spelling mistakes, please check the English.

Reviewer 3 Report

Report on the paper F-doped Ni-rich layered cathode material with improved rate 1 performance for lithium-ion batteries” by Zeng et al.

The paper deals about the effect of F doping in layered cathode materials for lithium-ion batteries. The system has been compared to the undoped material by using a various characterization technic (electronic microscopies, XRD, XPS) and elcetrochemical measurements, completed by a Density functional theory calculation. The results indicate that the rate performance of the F doped sample are improved the if the doping remains not too high. This is explained by an enlargement of the pore’s dimensions in the cathode, allowing a higher speed diffusion of the Li+ ions.

Considering the potential application, this subject is of course of interest for the scientific community. The procedure to produce the samples is correctly described and the measurements performed do not suffer any criticism. I have just the following remark the need to be corrected before publication ;

1)              I don’t think that a variation of 0.001 nm is detectable by electron microscopy. The evidence of a variation of the interplanar spacing (fig 6) should be weighted.

2)              The appearance of ring instead of peak on electronic diffraction does not indicates a polymorphism (line 225) but a disorientation of the planes.

3)              “With the increase of F doping amount, (….) the oxidation peaks around 4.0 V become less pronounced.” (lines 241-243) This is true for the F500 sample only but not obvious for the two other doped samples

4)              Typo

Lines 219-220 deal about the F500 sample (not the F0)

Round 2

Reviewer 2 Report

Accept in present form